

# Variation of choroidal thickness during the waking period over three consecutive days in different degrees of myopia and emmetropia using optical coherence tomography

Xianming Jiang[1,2,*], Ping Xiao[2,*], Qian Tan[3] and Yunxiao Zhu[2]

[1] Department of Ophthalmology, The Seventh Affiliated Hospital of Sun Yat-sen University, Shenzhen, Guangdong, China
[2] Health Management Center, The Seventh Affiliated Hospital of Sun Yat-sen University, Shenzhen, Guangdong, China
[3] Department of Ophthalmology, Shenzhen People's Hospital, Shenzhen, Guangdong, China
* These authors contributed equally to this work.

Corresponding author
Yunxiao Zhu,
zhuyunxiao@sysush.com

## ABSTRACT

**Purpose:** To investigate the diurnal variation in subfoveal choroidal thickness (SFCT) during the waking period over three consecutive days in different degrees of myopia and emmetropia.

**Methods:** A total of 60 adult volunteers were grouped into low, moderate, high myopia, and emmetropia subgroups. SFCT, axial length (AL), anterior chamber depth (ACD), and intraocular pressure (IOP) were measured every 2 h from 8 AM to 8 PM for three successive days.

**Results:** The mean values of daily change amplitude were 3.18 mmHg (IOP), 0.05 mm (AL), 0.17 mm (ACD), and 13.51 μm (SFCT). The values of AL and ACD increased simultaneously with spherical equivalent refraction (SER), but SFCT was the opposite. IOP had a diurnal variation, and there was no difference among the four groups. AL of the high myopia group, ACD of the emmetropia group, and SFCT of each myopia group had diurnal variation over three consecutive days. AL had a high mean value at noon every day, and SFCT had a low mean value at noon every day.

**Conclusion:** The choroid thickness of subjects with different degrees of myopia had a significant diurnal variation. The change of diurnal variation between emmetropic and myopic subjects may be one of the causes of myopia.

## INTRODUCTION

Myopia is the most common public health problem worldwide. It had been reported that choroidal thickness thinning was a feature of myopia occurrence and progression (*Fontaine et al., 2017*; *Jin et al., 2019*). Many researchers had studied the diurnal variation of choroidal thickness, but the results were inconsistent. Some studies reported there were

diurnal variations in choroidal thickness, (*Ahn et al., 2017*; *Chakraborty, Read & Collins, 2011*; *Lee et al., 2014*; *Tan et al., 2012*; *Usui et al., 2012*; *Zhao et al., 2016*) and while others reported not (*Burfield, Patel & Ostrin, 2018*; *Osmanbasoglu et al., 2013*; *Pollithy et al., 2015*). We found that the different results may be caused by the different spherical equivalent refraction (SER) of the research subjects. Considering the significance of the diurnal variation of choroidal thickness in myopia, especially in different degrees of myopia, we prospectively studied the diurnal variation of choroidal thickness and other ocular biometric parameters in adults with different degrees of myopia and emmetropia during the waking period of each day for three consecutive days. We aim to identify diurnal variations in choroidal thickness in the different degrees of myopic subjects.

# SUBJECTS AND METHODS

## Ethical approval

The study adhered to the tenet of the Helsinki Declaration. The study protocol was approved by the Research and Experimental Animal Ethics Committee of the Seventh Affiliated Hospital of Sun Yat-sen University, and the hospital's Institutional Review Board approval number was 2017SYSUSH-001.

## Subjects

This is a prospective, observational study performed in the Seventh Affiliated Hospital of Sun Yat-sen University. To fully cooperate with the study for three continuous days, we recruited 60 volunteers from hospital staff in the hospital. The inclusion criteria were 18–40 years of age, best-corrected visual acuity was ≥20/20, SER ranging was between +0.50 D and −10.00 D with astigmatism ≤−1.50 D, IOP ≤ 21 mmHg, and without retinal or choroidal abnormalities. The exclusion criteria included any ocular illnesses such as macular abnormality, glaucoma, previous ocular surgery or trauma, systemic vascular diseases such as hypertension and diabetes mellitus, menstrual women, and inability to cooperate during screening by OCT examination. The enrolled volunteers were allocated into four groups according to the SER. Emmetropia was defined as +0.50 D ≥ SER > −0.50 D, mild myopia was defined as −0.50 D ≥ SER > −3.00 D, and moderate myopia as −3.00 D ≥ SER > −6.00 D, and high myopia as −6.00 D ≥ SER > −10.00 D.

## Procedure

All participants underwent a complete ophthalmic examination before the study, including visual acuity, evaluation of the anterior segment with slit lamp biomicroscopy, fundus examination with direct ophthalmoscopy, and objective refraction. The refractive error was measured by an autorefractor (KR-8900; Topcon, Tokyo, Japan). Before the study began, each participant was informed that alcohol and caffeine were inhibited to maintain normal sleep during the research process. Optical parameters includin subfoveal choroidal thickness (SFCT), intraocular pressure (IOP), axial length (AL), and anterior chamber depth (ACD) were measured every 2 h from 8:00 to 20:00 for three consecutive days. Volunteers were engaged in normal work, life, and sleep at another time. IOP was determined by noncontact tonometry (CT-1; Topcon, Tokyo, Japan). Axial length and

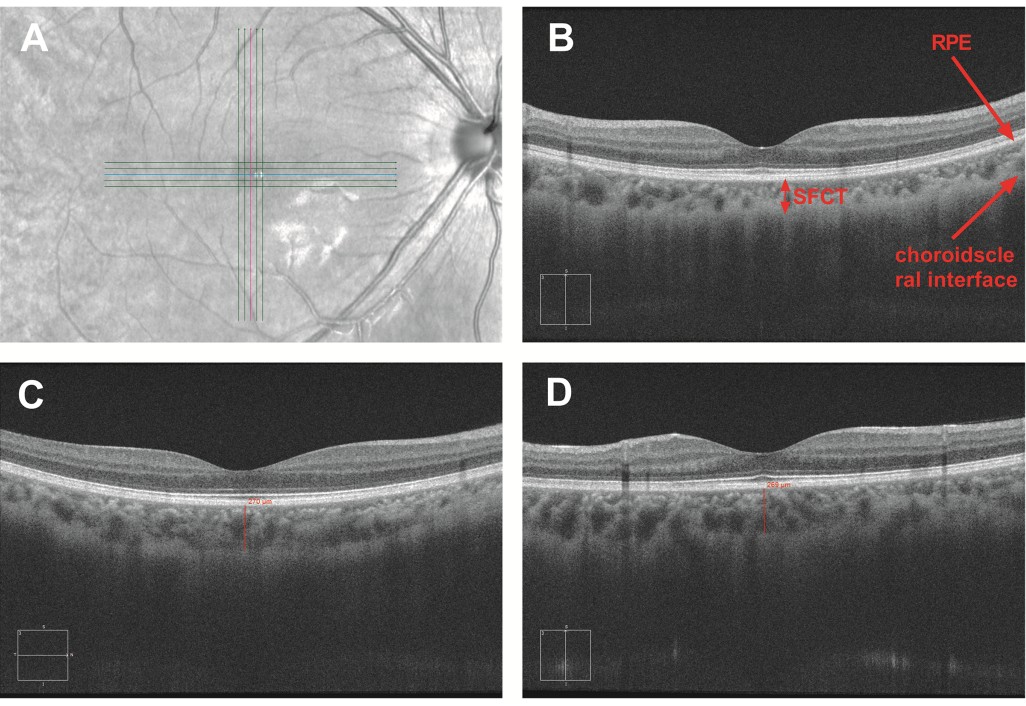

**Figure 1 Measurement of subfoveal choroidal thickness.** The SFCT value was obtained by cross-scan which could automatically uncover the foveal in horizontal and vertical orientations (A). SFCT value is defined as the vertical distance between the retinal pigment epithelium and the choroid sclera junction (B). SFCT value was measured in the horizontal and vertical directions (C and D).

anterior chamber depth were measured *via* interferometry (IOL-Master, Carl Zeiss, CA, USA). SFCT was obtained from spectral-domain OCT (HD-Cirrus OCT 5000; Carl Zeiss, CA, USA) with the enhanced depth imaging system. The HD-Cirrus OCT was a fast and safe method with a high scanning speed of 68,000 A-scans per second. The center wavelength of the OCT instrument was 840 nm, and the bandwidth was 90 nm. The axial resolution was five microns in tissue and 15 microns of transverse resolution. Adopting FastTracTM retinal-tracking technology allowed to make faster data acquisition and ensure that the effects of motion were significantly reduced (*Ang et al., 2019*). Cross-scan consisting of five horizontal and five vertical paralleled lines was used to obtain the choroidal images which could automatically uncover the foveal in horizontal and vertical orientations. The choroidal thickness was defined as the distance from the outer border of the retinal pigment epithelium (RPE) to the sclera border (*Ding et al., 2011*). To ensure the accuracy of data, the case was deleted if the choroid–scleral junction could not be visualized clearly. The mean SFCT was calculated as the mean of the horizontal and vertical subfoveal choroidal thickness measured independently by two experienced technicians (Fig. 1). Each technician measured each eye three times, taking the average value measured by two technicians as the final measurement of choroidal thickness.

**Table 1 Diurnal variation of ocular biometric parameters including IOP, AL, ACD, SFCT among different degrees of myopia and emmetropia groups on the first day.**

| Parameters | N | Variables | Amplitudes | P time | P time-SER | P SER |
|---|---|---|---|---|---|---|
| IOP (mmHg) | 60 | 15.67 ± 0.25 | 3.18 ± 0.14 | **0.000** | 0.359 | 0.599 |
| AL (mm) | 60 | 24.89 ± 1.22 | 0.05 ± 0.01 | 0.087 | 0.610 | **0.000** |
| ACD (mm) | 60 | 3.54 ± 0.03 | 0.17 ± 0.01 | **0.000** | 0.933 | **0.026** |
| SFCT (μm) | 60 | 250.00 ± 7.99 | 13.51 ± 1.19 | 0.158 | 0.262 | **0.000** |

Note:
Values and amplitudes represent mean ± standard error. Amplitudes are the means of the differences between the maximum and minimum of the first day. $P$ values are from repeated-measures ANOVA investigating the within-subjects effects of time and the time–SER interaction, the between-subjects effect of spherical equivalent refraction for the ocular biometric parameters on the first day. Significant $P$ values ($P < 0.05$) are highlighted in bold.

## Statistical analysis

We collected the binocular data of 60 adult volunteers, the data of right eyes were used for the following statistical analysis. The software used in the analysis was SPSS 22.0 and Graph Pad Prism 8.0. Analysis of variance with two factors (time and SER) was performed by repeated-measures analysis of variance on the data of the first day to identify any significant differences among the refractive error groups. The average change amplitudes (the difference between the maximum and minimum in 1 day) were also calculated. A repeated-measures analysis of variance with two factors (time and day of measurement) was performed to determine the significant diurnal changes in each of the parameters in each refractive error group. $P < 0.05$ was considered statistically significant and $P < 0.01$ was highly significant.

## RESULTS

The average age of participants was 26.71 ± 4.23 (range 21–37, $n = 60$), and the mean SER was −3.06 ± 2.51 D (range +0.50 to −8.50 D, $n = 60$). Data on the right eye were used in the following analysis. The emmetropic group consisted of 14 eyes, while the low, moderate and high myopic groups consisted of 15, 18, and 13 eyes, respectively. Spherical equivalent refraction was +0. 04 ± 0.40 D in the emmetropic group, −1.81 ± 0.52 D in the low myopic group, −3.77 ± 0.66 D in the moderate myopic group, and −6.79 ± 0.88 D in the high myopia group. The statistical results of seven-time points on the first day in Table 1 showed that IOP had diurnal variation ($P < 0.05$) and there was no difference between different spherical equivalent refraction groups ($P > 0.05$), other parameters (Al, ACD, SFCT) had the difference between different spherical equivalent refraction groups (All $P < 0.05$). There was no time–SER interaction in all parameters (All $P > 0.05$). This showed that time and SER were independent of each other.

According to SER, the data of parameters (AL, ACD, SFCT) were divided into four groups (emmetropia, low myopia, moderate myopia, and high myopia).
The repeated-measures ANOVA was carried out with the data of four groups on the first day, among which the results of multiple comparisons among groups were shown in Table 2. Table 3 and Fig. 2 showed the data for three consecutive days. Figure 2 is a line chart made by the groups with statistical differences in Table 3. The statistical results of seven-time points over 3 days showed that Al in the high myopia group, ACD in the

**Table 2 Multiple comparison of ocular biometric parameters including AL, ACD, SFCT among different degrees of myopia and emmetropia groups on the first day.**

| Parameters | Emmetropia | Low myopia | Moderate myopia | High myopia |
|---|---|---|---|---|
| $N$ | 14 | 15 | 18 | 13 |
| SER (D) | 0.04 ± 0.40 | −1.81 ± 0.52 | −3.77 ± 0.66 | −6.79 ± 0.88 |
| AL (mm) | 23.19 ± 0.214[a] | 23.97 ± 0.18[b] | 25.11 ± 0.20[c] | 26.16 ± 0.21[d] |
| ACD (mm) | 3.414 ± 0.067[a1] | 3.48 ± 0.057[a1] | 3.60 ± 0.062[b1] | 3.67 ± 0.065[b1] |
| SFCT (μm) | 304.18 ± 17.04[a2] | 295.39 ± 14.48[a2] | 235.43 ± 15.86[b2] | 165.01 ± 16.42[c2] |

Note:
Data of the groups are mean ± standard error. The results of multiple comparisons between groups are shown in superscripts. There is no difference between the two groups if the superscript letters are the same, and there is a difference if the superscript letters are different. $P < 0.05$ was considered statistically significant.

**Table 3 Diurnal variation of ocular biometric parameters including AL, ACD, SFCT in four groups among 3 days.**

| Parameters | Group | $N$ | Variables | Amplitude | Spherical equivalent refraction interaction, $P$ Value | Time of day effect, $P$ value | Day effect, $P$ value |
|---|---|---|---|---|---|---|---|
| AL (mm) | Emmetropia | 10 | 23.190 ± 0.107 | 0.04 ± 0.01 | 0.814 | 0.456 | 0.999 |
| | Low myopia | 10 | 23.973 ± 0.133 | 0.03 ± 0.00 | 0.108 | 0.082 | 1 |
| | Moderate myopia | 10 | 25.111 ± 0.110 | 0.05 ± 0.02 | 0.564 | 0.292 | 1 |
| | High myopia | 10 | 26.162 ± 0.086 | 0.05 ± 0.01 | **0.003** | 0.824 | 1 |
| ACD (mm) | Emmetropia | 10 | 3.414 ± 0.067 | 0.17 ± 0.02 | **0.000** | 0.846 | 0.893 |
| | Low myopia | 10 | 3.478 ± 0.057 | 0.15 ± 0.01 | 0.062 | 0.597 | 0.972 |
| | Moderate myopia | 10 | 3.600 ± 0.0662 | 0.16 ± 0.02 | 0.081 | 0.076 | 0.96 |
| | High myopia | 10 | 3.674 ± 0.065 | 0.19 ± 0.02 | 0.497 | 0.370 | 0.997 |
| SFCT (μm) | Emmetropia | 10 | 304.43 ± 13.80 | 17.82 ± 2.29 | 0.196 | 0.888 | 0.999 |
| | Low myopia | 10 | 296.21 ± 8.00 | 12.06 ± 0.74 | **0.000** | 0.918 | 0.991 |
| | Moderate myopia | 10 | 236.68 ± 7.09 | 10.88 ± 0.62 | **0.000** | 0.664 | 0.992 |
| | High myopia | 10 | 164.55 ± 7.87 | 10.98 ± 0.97 | **0.000** | 0.578 | 0.999 |

Note:
Data of the variables and amplitudes are mean ± standard error. Amplitudes are the means of changes over 3 days, and the changes are the differences between the maximum and minimum of each day. $P$ values are from repeated-measures ANOVA investigating the within-subjects effects of time and the time–day interaction, and the between-subjects effect of day for the ocular biometric parameters. Significant $P$ values ($P < 0.05$) are highlighted in bold.

emmetropia group, and SFCT in each myopia group had diurnal variation ($P < 0.05$). There was no difference between different days in all groups of all parameters (All $P > 0.05$). There was no time–day interaction in all groups of all parameters (All $P > 0.05$). This showed that time and day were independent of each other. In all groups of all parameters, there was no difference between the 2 days among 3 days.

# DISCUSSION

In this study, we investigated the diurnal variation of subfoveal choroidal thickness (SFCT), intraocular pressure (IOP), axial length (AL), and anterior chamber depth (ACD) in four groups with emmetropia, low myopia, moderate myopia, and high myopia during the waking period over three consecutive days. We found that the diurnal changes of Al, ACD, and SFCT were different among the four groups. IOP had a diurnal variation, and there was no difference among the four groups. The mean values of daily change amplitude

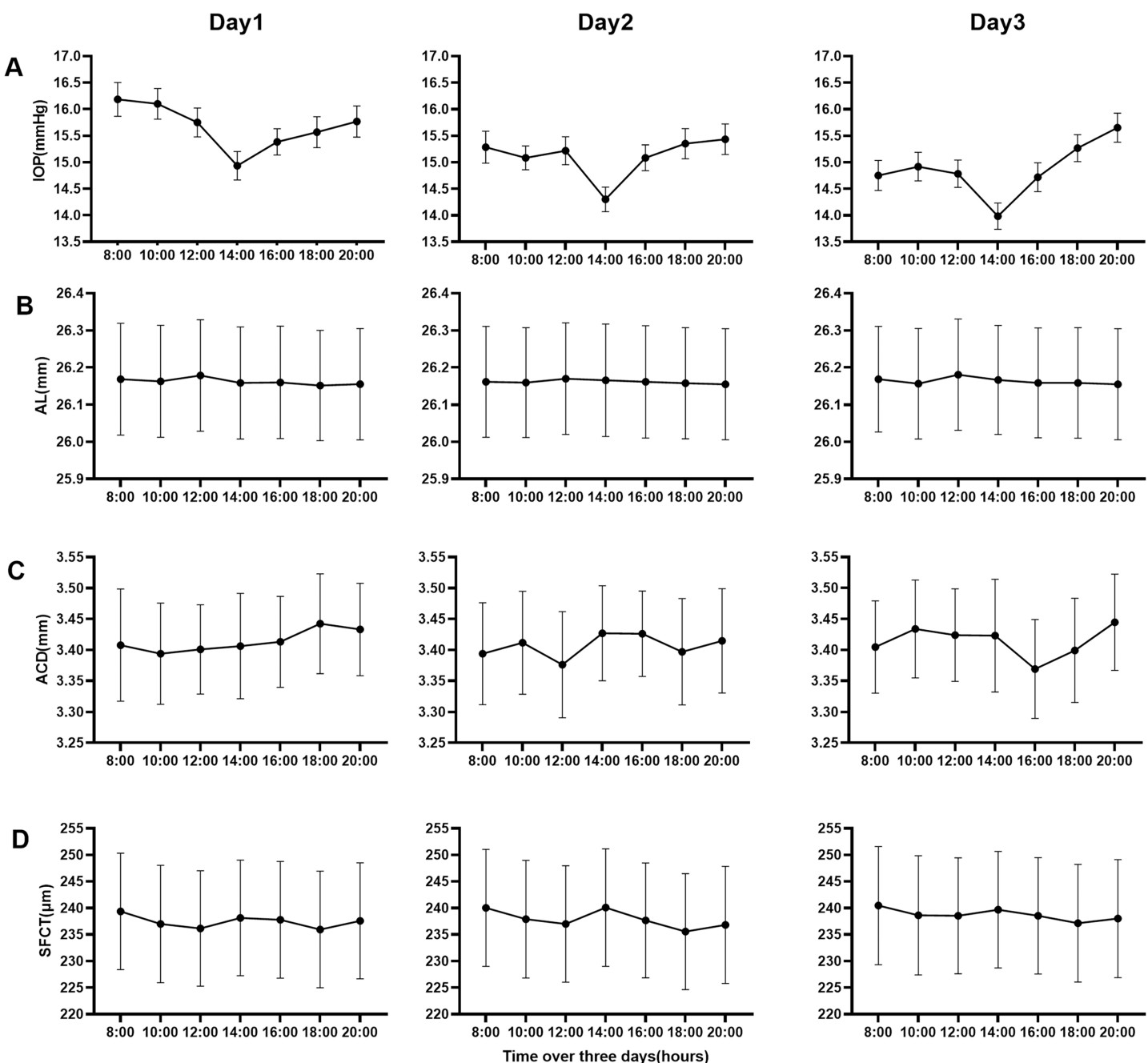

**Figure 2  Line charts of the mean of four parameters data over 3 days.** (A) Intraocular pressure of all groups, (B) the axial length of high myopia group, (C) anterior chamber depth of emmetropia group, and (D) the subfoveal choroidal thickness of all myopia groups.

of ocular biological parameters during the waking period on the first day were 3.18 mmHg (IOP), 0.05 mm (AL), 0.17 mm (ACD), and 13.51 μm (SFCT). The values of Al and ACD increased with the increase of SER, but SFCT was the opposite. AL in the high myopia group, ACD in the emmetropia group, and SFCT in each myopia group had diurnal variation over three consecutive days. The rhythm of variation during the waking period

on each of the 3 days was similar. IOP had a low mean value at 2:00 PM every day. Al had a high mean value at noon every day. On the contrary, SFCT had a low mean value at noon every day and another low mean value at 18:00.

The development of enhanced depth imaging (EDI) of spectral-domain optical coherence tomography (SD-OCT) with great sensitivity and repeatability made it possible to evaluate the choroidal thickness *in vivo* (*Branchini et al., 2012*; *Hoseini-Yazdi et al., 2019*; *Shao et al., 2013*; *Yamashita et al., 2012*). This provided excellent conditions for studying the diurnal variation of choroid thickness. However, the results of current studies on the diurnal variation of choroid thickness are inconsistent. The subjects in the studies which had found diurnal variations in the choroidal thickness were myopias. *Tan et al. (2012)* showed that a significant diurnal variation in CT was observed. Their mean axial lengths and spherical equivalents were 23.9 mm (range, 21.9 to 26.3 mm, SD ± 1.3 mm) and −0.46 D (range, −4.1 to +2.0, SD ± 1.3 D). Their sample size was only 12, which the emmetropia did not separate, and mainly reflected the result of myopia. *Usui et al. (2012)* showed that there was a significant circadian variation in SFCT and the mean SFCT was thinnest at 6:00 PM. In this study, most subjects were myopic, the mean refractive error was −4.4 ± 2.4 diopters and the mean AL was 25.4 ± 1.0 mm. The results on rhythm and trough time were similar to ours. *Lee et al. (2014)* showed similar results to those by *Tan et al. (2012)*. In this study, the mean refractive error was −3.27 ± 2.48 D, and the mean AL was 25.11 ± 1.43 mm (range, 22.43–29.82 mm). These values were also close to the values of the subjects of the myopic group in our study. *Zhao et al. (2016)* presented a very similar result with our low group. In this study, the refractive error of subjects was −2.00 (−3.75, −0.625) D (data presented as the median).

*Chakraborty, Read & Collins (2011)* measured 30 healthy individuals over two consecutive days and divided them into myopia and emmetropia groups according to SER. They found that there was no difference between myopia and emmetropia groups in intraocular pressure and choroidal thickness, and had a difference in axial length and anterior chamber depth. In addition to choroidal thickness, other results were similar to ours. The results of choroidal thickness were different from ours, which may be caused by different grouping criteria. Their grouping criteria were: emmetropes (SER, +0.75 to −0.75 DS) and myopes (SER, ≥ −1.00 DS). In this study, a Lenstar 900 was used to measure SFCT, which was rarely used in the current study. Besides, it must use data from a single day to investigate the between-subjects effect of spherical equivalent refraction in repeated-measures ANOVA, which was not explained in the author's article about this. *Burfield, Patel & Ostrin (2018)* found that there was no difference between myopia and emmetropia groups in choroidal thickness, axial length, and anterior chamber depth. Compared with Chakraborty's results, the same grouping criteria were used, and Al and ACD were measured using the same measuring equipment Lenstar 900, but the results of Al and ACD were opposite. In Burfield's study, the choroidal thickness was choroid thickness in the central 1-mm region, not choroidal thickness in the subfoveal, the time points they measured were included in the sleep period (4:00), so the results on the choroidal thickness may not be comparable. But both these two studies found that the choroidal thickness was thinnest at noon, and our results also showed a low choroidal

thickness at noon. In many studies, there were no diurnal changes in choroid thickness, because the subjects they selected were emmetropic eyes. *Osmanbasoglu et al. (2013)* found the mean SFCT in all groups did not show significant variation during working hours. A total of 85 subjects in this study had a mean refractive error of +0.50 ± 0.25 diopters and a mean axial length of 23.1 ± 0.8 mm. These values were close to the values of the subjects of the emmetropic group in our study. In the study by *Pollithy et al. (2015)*, a significant diurnal variation of choroidal thickness wasn't observed during the waking period. Subjects in this study had an average spherical equivalent of +0.56 ± 0.96 DPT and an average axial length of 23.3 ± 0.97 mm. These values were close to the values of the subjects of the emmetropic group in our study. The research method, measurement period, and times were similar to our study. The diurnal variation of SFCT was indeed not found in our study's emmetropic group, similarly. *Baek et al. (2018)* found that the diurnal macular choroidal thickness values measured in the healthy subjects and POAG patients were not statistically significant during the daytime. The healthy subjects had a mean spherical equivalent of −2.51 ± 3.31 diopters. The large standard deviation of the spherical equivalent of this group data may lead to the poor representativeness of the mean. The healthy subjects may not necessarily be myopia.

Therefore, our conclusion could unify the current research results on the diurnal variation in subfoveal choroidal thickness. When studying 24-h circadian changes, participants were woken up from sleep, which destroys the original circadian rhythm. So, most of the current research on diurnal variation was the changes of 12 h during the awake period. We believed that the 12-h diurnal variation did not represent the circadian rhythm of the biological clock, but only showed the oscillation of these parameters in the daytime. In our study, the diurnal changes in the ACD emmetropia group were statistically significant, which may indicate that the lens position of the emmetropia group changed after waking. With the occurrence and development of myopia, although ACD gradually deepens, the oscillation was not obvious, which may indicate that the oscillation disappeared when the position of the lens moved backward to a certain position after the formation and deepening of myopia. This suggested that the oscillation of lens position might be a protective factor against myopia, which needed further study. Similarly, based on our results, we speculated that the subfoveal choroidal thickness was stable during the 12-h waking time in emmetropia, but there was an oscillation in the subfoveal choroidal thickness in myopia. The possible reason was that, after the formation of myopia, the blood vessels of the choroid had changed, which led to the concussion of the filling state of blood vessels. *Kinoshita et al. (2017)* found that the main reason for the formation of the daily variation of the choroid was the change in the choroidal vascular caliber. There was an oscillation in the axial length in the high myopic group, and the oscillation only appeared in high myopia. High myopia may result in vision impairment caused by a retinal detachment, myopic macular degeneration, chorioretinal atrophy, choroidal neovascularization, and macular schisis (*Barteselli et al., 2014*; *Chang et al., 2013*; *Koh et al., 2016*; *Ruiz-Medrano et al., 2019*). Whether the oscillation was related to the complications of high myopia needs further study.

The amplitude and phase of the diurnal variations in choroidal thickness were significantly altered with the myopic defocus (*Chakraborty, Read & Collins, 2012*). *Kearney et al. (2017)* showed that the serum melatonin concentration in the morning of young adults with myopia was significantly higher than that of no myopia. The shift in the circadian phase was caused by the increase in melatonin concentration (*Ostrin, Abbott & Queener, 2017*). Melatonin resynchronized the circadian rhythm (*Satyanarayanan et al., 2018*). Melatonin changes the phase of the circadian rhythm (*Crowley & Eastman, 2013*). According to these results, we speculated that the diurnal variations of choroidal thickness may change after myopia, which also confirmed that our emmetropic and myopic patients had different results in the diurnal change of choroidal thickness. The diurnal variation of choroidal thickness was closely related to myopia. *Loman et al. (2002)* reported that the disturbance of the daily light-dark period may also lead to human refractive error. Now a large number of studies have confirmed that outdoor activities can prevent myopia (*He et al., 2015*; *Zadnik & Mutti, 2019*). The change in diurnal variation of choroidal thickness may also be an initiating factor of myopia. This needs further study.

There were several limitations in this study. First, the measurements of choroidal thickness were performed manually by two examiners, measuring errors from measurement software should be considered further. Second, 60 participants enrolled in this study can only represent adults from 20–40 years old and the refractive status of hypermetropia was not considered. Third, we only measured the diurnal changes of SFCT and other parameters from 8:00 to 18:00, but the data during the night were absent. We only demonstrated the phenomenon of diurnal change of SFCT, further studies to explore the mechanism will be needed.

## CONCLUSION

Significant diurnal variations in choroidal thickness in the different degrees of myopic subjects were found, but not in the emmetropic subjects, and SFCT had a low mean value at noon every day. In clinical practice and research on choroidal thickness, attention should be paid to the importance of examination time, especially in patients with myopia. The diurnal variation of choroidal thickness was different in emmetropia and myopia. The change of diurnal variation may be one of the initiating factors of myopia.

### Funding
The authors received no funding for this work.

### Competing Interests
The authors declare that they have no competing interests.

### Author Contributions
- Xianming Jiang conceived and designed the experiments, performed the experiments, analyzed the data, prepared figures and/or tables, authored or reviewed drafts of the article, and approved the final draft.

- Ping Xiao conceived and designed the experiments, authored or reviewed drafts of the article, and approved the final draft.
- Qian Tan analyzed the data, authored or reviewed drafts of the article, and approved the final draft.
- Yunxiao Zhu analyzed the data, authored or reviewed drafts of the article, and approved the final draft.

## Data Availability

The raw data is available in the Supplemental File.

## Supplemental Information

Supplemental information for this article can be found online at http://dx.doi.org/10.7717/peerj.15317#supplemental-information.

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
