# Peer review of "Variation of choroidal thickness during the waking period over three consecutive days in different degrees of myopia and emmetropia using optical coherence tomography"

_PeerJ, doi:10.7717/peerj.15317_

## Round 0.1 · original submission · Minor Revisions

The reviewers were positive about aspects of the report, but also express some reservations about your manuscript and request that additional work be done to attempt to address these concerns. If you believe that your manuscript can be adequately revised to answer these points, I will be happy to consider a revised version.

Reviewer 1 ·

Basic reporting

Well-written manuscript, with good technical standard, clear hypothesis answered by results and conclusions. Some minor improvements could be made:
Ln 43, ln 58, ln 70-71 – please expand those abbreviations. Readers who are not ophthalmologists should be able to understand the manuscript.
Some language amendments are required. Please rephrase following sentence for better text clarity:
Ln 46-47 – the aim of the study should be clearly stated
Ln 156-157 – difficult to understand
Ln 167 – improper language
In Results section was used weird P reporting style. Please replace all those “Ptime”, “PSER”, “P Time-SER” with simple “P”.
Ln 145 – please correct the citation format.

Experimental design

Ln 86 – give rationale for this method
In “Statistical Analysis” subsection – How did authors present the data? As median ± IQR or mean ± SD?
Ln 94-95 – Why did authors divide associations for significant and highly significant? This division was not necessary in my opinion.
Ln 99 – “Data on the right eye were used in the following analysis” – please rephrase and place this sentence in relevant “Methods” subsection.

Validity of the findings

Results are consistent with current literature. Study adds some new data. Conclusions well stated.

Reviewer 2 ·

Basic reporting

Manuscript has a proper scientific structure, written in unambiguous English. However, some proof reading should be done. Aim of study was awkwardly phrased and should be rewritten (ln 46-47). Authors did not expand utilized abbreviations in the text (methods section). Please use standard P reporting style instead of “P time” etc.

Experimental design

Please describe how many eyes the authors examined and indicate in methods section what eye did you select (or both).

Validity of the findings

Good manuscript with relevant findings, self-contained hypothesis. Aim of the study was answered by the results and the conclusions. Accept after proof-reading and minor improvements.

Reviewer 3 ·

Basic reporting

Language & structure:
The manuscript has a proper scientific structure, proficient English but some improvements should be made:
Ln 24 – please expand this abbreviation
Ln 39-42 – please rephrase this complex sentence for better clarity
Ln 43 – what is SER? Please expand this abbreviation
Ln 46-47 – please rephrase. “Hope” is wrong word for the aim of study
Ln 70 – SFCT – please expand this abbreviation
Ln 81-82 – “The choroidal thickness..” – please refer to relevant literature
Ln 95 – Why did you use two separate cut-offs for P value? There is no reason behind it.
Ln 104-107 & Ln 114-117 – Please report P values in the standard form
Ln 117 – please report those P values
Ln 121-123 – please give abbreviations in brackets
Ln 186-188 – please rephrase this sentence
Figure 2 – please report P values on those graphs.

Experimental design

There is good experimental design, no flaws were found.

Validity of the findings

Findings are in accordance with the current literature and statistically sound. Suitable for publication.

---

## Round 0.2 · accepted · Accept

I have now had the opportunity to read your revised manuscript, and your responses to the reviewers' comments. I believe that you have addressed the concerns raised, and I am happy to accept your manuscript.

Reviewer 1 ·

Basic reporting

English is clear, manuscript is easy to understand. Sufficient background was provided. Authors responded to my previous comments. Tables & figures are clear.

Experimental design

Methods described sufficient to replicate. No significant flaws in the methodology were noted. Investigation was performed with technical and ethical standards. Research question was well stated and answered by the results and conclusions.

Validity of the findings

Conclusions well stated, supported by the results and the discussion.

Reviewer 2 ·

Basic reporting

Tables and figures are clear. English is clear, manuscript is easy to understand. Sufficient background was provided.

Experimental design

Methods do not have significant flaws, are described sufficient to replicate. Well stated research question answered by the results. No ethical or technical issues were found.

Validity of the findings

Well stated conclusions that are in line with the results.

Reviewer 3 ·

Basic reporting

Authors responded well to the previous comments. English is unambiguous and coherent. Sufficient background was provided

Experimental design

Methods do not have significant flaws, are described sufficient to replicate. Well stated research question answered by the results. No ethical or technical issues were found.

Validity of the findings

Well stated conclusions that are in line with the results.